# Pathogenetic Mechanisms Underlying Major Adverse Cardiac Events in Personality Type D Patients after Percutaneous Coronary Intervention: The Roles of Cognitive Appraisal and Coping Strategies

**DOI:** 10.3390/diagnostics13213374

**Published:** 2023-11-02

**Authors:** Alexey N. Sumin, Anna V. Shcheglova

**Affiliations:** Laboratory of Comorbidity in Cardiovascular Diseases, Department of Clinical Cardiology, Research Institute of Complex Problems of Cardiovascular Diseases, 6, Sosnovy Blvd., 650002 Kemerovo, Russia; nura.karpovitch@yandex.ru

**Keywords:** type D personality, pathogenetic mechanisms, coping style, cognitive appraisal, coronary artery disease, prognosis, percutaneous coronary intervention

## Abstract

Background: This paper aimed to study the association of type D personality, coping strategies, and cognitive appraisal with annual prognosis after a percutaneous coronary intervention (PCI) in patients with coronary artery disease (CAD). Methods: The prospective study included 111 CAD patients who underwent a PCI. All participants, before the PCI, completed questionnaires designed to collect information about type D personality, cognitive appraisal, and coping styles. Information was also collected on the clinical and demographic characteristics of the patients. After 1 year of follow-up, the presence of major adverse cardiac events (MACEs) was assessed. Results: The presence of a MACE was noted in 38 patients, and the absence of a MACE was noted in 53 patients. In patients with type D personality, higher incidences of MACEs (54.1% versus 33.3%; *p* = 0.0489) and hospitalization rates (29.7% versus 7.4%; *p* = 0.004) were revealed. Patients with poor prognoses preferred a moderate use of the confrontation strategy than patients without a MACE (78.4% vs. 50.9%; *p* = 0.0082). Patients with MACEs had statistically significantly lower indicators of strong emotions (11.92 ± 5.32 versus 14.62 ± 4.83 points; *p* = 0.005) and future prospects (11.36 ± 3.81 versus 13.21 ± 3.41 points; *p* = 0.015) than patients without a MACE. In a multiple binary logistic regression model, the following factors had significant associations with MACE development: type D, moderate use of confrontation coping, moderate use of self-control coping, and strong emotions in cognitive appraisal. Conclusion: This study showed that not only personality type D, but also certain coping strategies and cognitive appraisals increase the likelihood of developing a MACE after a PCI. This provides a theoretical basis for understanding the mechanism underlying type D personality and MACEs in patients after a PCI.

## 1. Introduction

Personality type D (“distressed”) is characterized by a combination of negative affectivity (NA) and social inhibition (SI) [1]. NA is characterized by a tendency to experience many negative emotions and combines feelings of dysphoria, anxiety, and irritability. SI is manifested by a conscious tendency to suppress the expression of emotions in social interactions due to social discomfort, inhibition, and a lack of social balance [1]. The D personality type is thought to reflect the synergistic effects of NA and SI, with the conscious suppression of these negative emotions leading to adverse health outcomes (primarily cardiovascular health). So, the presence of this personality type has a negative impact on the prognoses of patients with coronary artery disease (CAD). This was shown in the initial studies of personality type D [2,3]. Although not all researchers have confirmed the presence of a predictive effect for patients with personality type D [4,5], a meta-analysis has shown that type D personality predicts a two-fold increase in the risk of mortality among cardiac patients [6]. To date, studies continue to link type D personality with a poor prognosis in cardiovascular disease [7,8,9]. Also, a recently published meta-analysis of 19 prospective cohort studies showed that type D personality predicts adverse events in patients with CAD [10].

Further research has focused on elucidating the pathways that contribute to the relationship between type D personality and adverse cardiovascular health. Indirect mechanisms (behavioral) are primarily related to the role of health behavior: patients with type D personality consume more junk food, exercise less, and make less effort to control their weight compared to patients without type D personality [11], and they have poor adherence to medications/treatment [12,13]. Patients with higher scores on type D personality subscales had a lower need for information about psychological well-being in cardiac rehabilitation programs [14]. It should also be taken into account that personality type D may be an independent risk factor for the development of moderate cognitive dysfunctions, which has been shown, in particular, in patients with arterial hypertension [15]. A study by Buczkowska M et al. [16] showed that type D personality increases the risk of poor health behavior by more than five times. In addition, patients with a distressed personality have been shown to exhibit the least effective psychological attitude and the least effective preventive behavior and differ significantly in this respect from other personality types (intermediate and non-type D). Direct mechanisms (biological) have been primarily associated with the influence of personality type D on various physiological processes regulated by neurohormonal systems (for example, the hypothalamic–pituitary axis and the sympathetic–adrenal–medullary axis) and the autonomic nervous system accompanying prolonged psychological distress [17].

Since it has been shown that personality type D has a higher perception of stress in relation to life events, the response of the cardiovascular system to acute psychological stress has attracted considerable attention from researchers. The cardiovascular reactivity hypothesis states that prolonged or enhanced cardiovascular responses (e.g., blood pressure and heart rate) to psychological stress are predictors of future risk of cardiovascular disease. Prospective studies have shown that not only exaggerated cardiovascular responses to psychological stress associated with the development of adverse outcomes [18], but also an atypically low (i.e., blunted) cardiovascular response to psychological stress [19].

Personality type D is associated with both high [20] and low [21] cardiovascular responses to acute psychological stress. It is believed that it is the attitude of the subject with personality type D to stress exposure that explains the reaction of the cardiovascular system to stress; the reaction is increased with stressors of high social significance and reduced with stressors of low social significance [22].

One of the tasks in the treatment of CAD patients with personality type D is the modification of the patients’ reactions to the stressful effects of everyday life [23]. Therefore, it is of interest to study the possible pathogenetic significance of their cognitive assessments of stress and strategies for coping with stress. For example, among the adequate strategies for overcoming stress are “Planful Problem-Solving”, “Self-Control”, “Positive Appraisal”, and “Accepting Responsibility”. The following coping strategies are less adequate: “Confrontative Coping”, “Seeking Social Support”, and, especially, “Distancing and Escape/Avoidance” [24]. Previously, it was shown that maladaptive coping strategies prevail in people with personality type D, both in healthy individuals [24] and in people with various diseases [25], particularly in CAD patients [8,26,27]. Moreover, a recent study showed that inadequate coping strategies mediate the adverse effects of personality type D on the prognoses of CAD patients after a PCI [8].

It has been previously shown that coping ability is affected by cognitive appraisal. There is also known variability in coping depending on whether stress is perceived as a loss, threat, or challenge [28,29]. Therefore, it is necessary to study the internal prerequisites that determine the actualization of coping strategies, which is solved when developing the concept of the cognitive appraisal of stress [30] or a difficult situation [31]. There are not many studies on the cognitive appraisal of stressful situations in CAD patients. Nevertheless, in a study by Lv et al. [8], it was shown that cognitive appraisal (threat appraisal and challenge appraisal) mediated the influence of personality type D on prognosis.

However, it turned out that this study noted the cultural specificity of coping strategies in Chinese CAD patients; the “Acceptance-Resignation” strategy prevailed. At the same time, in other regions, other inadequate coping strategies were identified for type D personality; for example, in Russian studies, the escape–avoidance strategy was noted [32,33]. Therefore, there was a need to further study this issue in other populations. This served as the basis for conducting this study in one of the regions of Russia, in the south of Western Siberia. The aim of this study was to study the influences of personality type D, coping strategies, and cognitive appraisal on the annual prognosis after a PCI in patients with coronary artery disease.

## 2. Materials and Methods

### Patients and Procedures

The prospective study included 260 patients aged 33 to 81 years who were consecutively admitted to prepare for an elective percutaneous coronary intervention (PCI) at the Research Institute for Complex Problems of Cardiovascular Diseases (Kemerovo) from December 2020 to October 2021. The study protocol was approved by the local Ethics Committee of the institution. All patients filled out informed consent forms to participate in the study. The inclusion criteria for the study were stable coronary artery disease requiring endovascular intervention and the ability to complete the questionnaire. The exclusion criteria were acute coronary syndrome, severe comorbid background, and refusal of the patient to participate in the study.

After taking into account the inclusion and exclusion criteria, the observation group consisted of 112 patients. At the pre-hospital stage, all patients underwent a standard preoperative examination. To analyze the anamnesis, clinical, and instrumental information of the patients, the patients’ medical records were studied. Among the clinical indicators, age, genders of patients, the presence of risk factors, concomitant diseases, a history of stroke, myocardial infarction, and data from instrumental examination methods were taken into account (Figure 1).

## 3. Measures

### 3.1. Type D Personality

To determine whether a patient had personality type D, we used the DS-14 questionnaire, including the NA (“negative affectivity”) and SI (“social inhibition”) subscales, which were translated into Russian and adapted and validated by Pushkarev G et al. [34]. In this study, the Cronbach’s alpha for NA was 0.78, and for SI, it was 0.74, which confirms the adequacy of the intrinsic structure of the Russian version of DS14. The questionnaire contains 14 multiple choice questions with the following answer options: incorrect, rather incorrect, difficult to say, perhaps true, and absolutely true. Each answer has its own score; if there are 10 points or more on the NA and SI scales, type D personality is established. Not only was type D personality analyzed as a dichotomous variable, but also the z-scores for NA and SI and their statistical interactions were assessed.

### 3.2. Hospital Anxiety and Depression Scale (HADS) Personality

The Hospital Anxiety and Depression Scale (HADS) translated and adapted to the Russian language was used to assess the level of depression and anxiety. The scale is made up of 14 statements serving 2 subscales: subscale A (“anxiety”) with odd items (1, 3, 5, 7, 9, 11, and 13) and subscale D (“depression”) with even points (2, 4, 6, 8, 10, 12, and 14). Each statement corresponds to 4 answer options, reflecting gradations of symptom severity and coded according to the increasing severity of the symptom from 0 points (absence) to 4 points (maximum severity). The higher the overall score, the more pronounced the symptoms of anxiety or depression. In the presence of 8–10 points, subclinically expressed anxiety/depression was determined; at 11 points and above, clinically expressed anxiety/depression was determined.

### 3.3. Coping Style

Coping strategies were assessed using the Ways of Coping Questionnaire (WCQ). The questionnaire was developed by Folkman S and Lazarus R, and was adapted into Russian by L.I. Wasserman et al. [35]. The questionnaire contains fifty different behaviors in a problematic or difficult life situation. Depending on how often the subject uses the described behavior, they are offered statements (“never”, “rarely”, “sometimes”, and “often”). These statements are evaluated on a 4-point system and are grouped into the following scales: confrontational coping, problem solving and planning, self-control, positive reappraisal, accepting responsibility, distancing, seeking social support, and escape–avoidance. Confrontational coping involves aggressive behavior in order to change the situation, hostility, and a willingness to take risks. The coping strategy of problem solving and planning is characterized by activities that include analysis and the development of algorithms to solve the problem. The coping strategy of self-control includes inherent efforts to regulate and manage one’s emotions and actions. For the positive reappraisal coping strategy, positive reassessment is inherent in efforts to find positive moments in a problematic or difficult life situation. The coping strategy of accepting responsibility is the awareness of one’s own role in the emergence of a problem and the ability to find possible ways to solve it. The coping strategy of distancing involves efforts to separate oneself from the problem situation and reduce its significance. The coping strategy of seeking social support involves asking others for help. Escape–avoidance coping strategies are characterized by efforts that are aimed at avoiding a difficult life situation. The processing of “raw” indicators was carried out by converting the scores to standard T-scores. In addition, the degree of severity of one or another coping strategy in a patient was defined as rare use, moderate use, or a pronounced preference for the corresponding strategy.

### 3.4. Cognitive Appraisal

The subjective assessment of a difficult life situation was determined using the author’s methodology, “Appraisal Criteria of the Life Situation Difficulty” by E. V. Bityutskaya [36]. The methodology consists of two parts. In the first one, the respondent is asked to list and briefly describe situations in their own life that they perceive as difficult. The second part presents 34 evaluation statements. The subject is asked to correlate the situations described in the first part with each statement and put down the appropriate marks using a 7-point scale (from 0—“no, totally wrong” to 6—“yes, absolutely right”). The results reveal why the respondent finds the situation difficult. As a result, the subjective assessment of the situation is represented by the following scales: (1) general features of difficult situations; (2) uncontrollability of the situation; (3) unclearness (ambiguity) of the situation; (4) the need for a quick and active response; (5) difficulty of making a decision (dilemma); (6) difficulty of predicting the situation; (7) negative emotions; and (8) threat for the future. The following fit indices of the questionnaire were obtained: RMSEA = 0.044; CFI = 0.910; χ^2^ = 912.899; and df = 378 [31].

Orientations in a difficult situation were measured using the situational version of the “Types of Orientations in Difficult Situation” (TODS; [37]) questionnaire. Patients rated 76 statements grouped into 38 pairs on a scale from 0 to 3 (0—“totally wrong”; 1—“somewhat wrong”; 2—“somewhat right”; and 3—“absolutely right”). The calculation of the indicator of each orientation in difficult situations was carried out by finding the average value on the scale. The questionnaire makes it possible to diagnose eight orientations: five orientations characterize the subject’s efforts to approach difficulties (desire for difficulties = drive; focus on high labor intensity = thoroughness, focus on threat signals = threat alert; opportunity orientation; obstacle orientation), and three orientations involve avoiding difficulties (loss orientation; difficulty avoidance = rejection; resource conservation orientation = inaction; orientation towards ignoring difficulties = insouciance). The questionnaire’s structural model fits well: RMSEA = 0.049, CFI = 0.900, χ^2^ = 3068.835, df = 1171 [31].

### 3.5. Follow-Up Observation

The long-term results of surgical intervention on the coronary arteries were assessed, on average, after a 1-year follow-up using active telephone monitoring. It was possible to collect information about the state of health of 91 (81.3%) patients. We compared the patients who dropped out of the study and those presented in the article. The results of this comparison showed the comparability of these groups in terms of initial indicators (Appendix A). In the annual period, the following MACEs were analyzed: cardiovascular and non-cardiovascular death, myocardial infarction, acute cerebrovascular accident/transient ischemic attack, repeated coronary angiography and/or percutaneous coronary intervention, resumption of the angina pectoris, and hospitalization for cardiovascular diseases. For analysis, two groups were formed: a group without MACEs (*n* = 53) and a group with MACEs (*n* = 38) (Figure 1).

### 3.6. Statistical Analyses

Statistical processing was carried out using the SPSS 17.0 software package. The distribution of quantitative variables was tested for normality using the Kolmogorov–Smirnov test. With a normal distribution, the data were presented as means (M) and standard error of the mean (SE), with the non-normal distribution presented as a median (Me) and quartiles (25th and 75th percentiles). Student’s *t*-test, the Mann–Whitney test, and the chi-squared test were used to compare the two groups. To assess the factors associated with the presence of an unfavorable prognosis after a PCI, a binary logistic regression analysis (forward stepwise LR method) was carried out; the following variables were included in the model: risk factors, clinical characteristics, and questionnaire data (DS-14—Type D, Zscore NA, Zscore SI, ZNA × ZSI, WCQ questionnaire, Appraisal Criteria of the Life Situation Difficulty, and TODS). Performance of type D parameters in discriminating the risk of an unfavorable prognosis (MACE development) after a PCI was evaluated through receiver operating characteristic curve analysis. The level of critical significance (*p*) was taken as 0.05.

## 4. Results

The pattern of MACE development is shown in Figure 2. A total of 38 patients (20 men and 18 women) underwent a MACE within 1 year after a PCI, including cardiac death (3.0%), stroke (3.0%), recurrent angina (40.0%), repeated CAG and/or percutaneous coronary intervention (14.0), and hospitalizations for cardiovascular diseases (40.0%).

The presence of personality type D was detected in 37 patients, and the absence of personality type D was detected in 54 patients. When analyzing cardiovascular events, a significantly higher frequency was revealed in MACEs in general (54.1% versus 32.1%; *p* = 0.049) and in the frequency of hospitalizations in particular (29.7% versus 7.4%; *p* = 0.004) in the group of patients with type personality D compared with the individuals who were not related to type D personality. Otherwise, the groups did not differ in the frequency of other adverse events (Figure 3).

When analyzing the initial clinical and anamnestic data in the groups with the presence (*n* = 38) and the absence of MACEs (*n* = 53), it was noted that the groups of patients were comparable in all respects (Table 1). More than half of the patients in both groups had a previous myocardial infarction, with a single coronary artery lesion predominating, both in the group without MACEs (75.5%) and in the group with MACEs (68.4%, *p* > 0.05).

The mean scores on the subscales of the DS-14, HADS, and WCQ questionnaires among the entire cohort of those examined are presented in Table 2. The frequency of personality type D being identified was higher among patients with MACEs compared with patients without MACEs (*p* = 0.049). The other indicators of these questionnaires did not differ significantly; however, in the MACE group, there was a trend towards higher rates of negative affectivity, social inhibition, anxiety, and depression. The mean values of the subscales of coping strategies also did not differ on any scale in the groups with/without MACEs. With a more detailed assessment of coping strategies, highlighting the frequency of using one or another coping strategy, differences have already been identified (Figure 4). Thus, patients in the group with a favorable prognosis more often used a rare use of the confrontation strategy (without MACEs, 24.5% versus with MACEs, 2.7%; *p* = 0.0049), and the patients with unfavorable prognoses preferred a moderate use of the confrontation strategy (with MACEs, 78.4% versus without MACEs, 50.9%; *p* = 0.0082) according to the WSQ.

An analysis of the subjective assessment of a difficult life situation demonstrated high scores on the “Common signs of difficult life situations” scale in all participants, with no statistically significant difference between patients with or without MACEs (Table 3). The leading cognitive appraisal of the situation difficulty in the groups was the need for a quick response (16.2 ± 4.96 points in patients without MACEs; 15.81 ± 4.61 points in patients with MACEs). The patients with MACEs had statistically significantly lower indicators in the “Strong emotions” (11.92 ± 5.32 points vs. 14.62 ± 4.83 points, respectively; *p* = 0.005) and “Future prospects” (11.36 ± 3.81 points versus 13.21 ± 3.41 points, respectively; *p* = 0.015) scales compared to the patients without MACEs. With a more detailed cognitive appraisal in the TODS questionnaire in the studied groups, no significant difference was achieved. Only the tendency towards a greater focus on the resource conservation orientation (i.e., inaction) in the group with the subsequent development of MACEs (*p* = 0.094) involuntarily attracts attention (Table 4).

In a multiple binary logistic regression model (forward LR method), the following factors had a significant association (χ^2^(4) = 22.98; *p* = 0.001) with MACE development: type D personality (B = 1.381; *p* = 0.025), moderate use of confrontation coping (B = 1.641; *p* = 0.011), moderate use of self-control coping (B = 1.343; *p* = 0.039), and strong emotions in the ACLSD questionnaire (B = −0.132; *p* = 0.040). This model explained only 39.1% (Nagelkerke R2) of the variance in MACEs and correctly classified 79.1% of cases (Table 5).

Among the psychological factors of the construct of personality type D (Type D, NA, SI, and ZNA × ZSI), the greatest association with the development of adverse events during the year was noted for personality type D as a dichotomous variable. At the same time, no association with MACEs was found separately for the NA or SI scales or their interaction (Table 6).

As shown in Table 7 and Figure 5, the largest area under the curve was for the ZNA × ZSI indicator, reflecting the possible synergistic effect of the negative affectivity and social inhibition subscales. However, the area for the curve for this variable was <0.7, indicating unacceptable discrimination.

## 5. Discussion

The present study showed that in patients with CAD with personality type D, in contrast to patients with non-type D personality, during the year after the PCI, MACEs and hospitalizations for cardiovascular diseases were more likely to develop. In the group with unfavorable prognoses, in addition to the more frequent occurrence of personality type D, there was a greater preference for moderate use of the confrontational coping strategy. On cognitive appraisal, patients with MACEs had statistically significantly lower indicators of strong emotions and future prospects than patients without MACEs.

In previous studies of coping strategies in individuals with personality type D, as a rule, maladaptive ways of coping with stress were identified. Thus, among young and healthy individuals, it was shown that the escape–avoidance coping strategy was associated with personality type D, and the positive reappraisal strategy was associated with non-type D personality [32]. Patients with coronary artery disease also showed similar indicators; the coping strategies of “Avoidance”, “Acceptance-humility” [8], and “Shifting responsibility” [27] were more common, and the “Confrontation” [8] and “Planning” [27] coping strategies were used less often. Differences in the severity and specific manifestations of inadequate coping strategies in this study and in previous studies can be explained both by the influences of ethnic and cultural characteristics and by the influences of age and the presence of cardiovascular disease. For example, for Chinese residents, the traditional attitude of “feel at ease under any circumstances” may encourage the use of the “Acceptance-Humility” strategy among Chinese patients. Accordingly, patients with type D personality define the disease as a stressful uncontrollable situation, they feel hopeless, and they tend to perceive the disease as the end of life. Therefore, they are reluctant to seek treatment and support, which may explain their poor adherence to treatment [38].

Despite the rather close attention that is paid to the coping strategies of type D personality, their pathophysiological mechanisms of influence on the prognosis have not been sufficiently studied to date. We can only recall the study by Lv et al. [8] that has a design that is similar to our study. In this work, it was shown that personality type D had both direct and indirect negative impacts on the frequency of MACE development through the acceptance-resignation coping strategy. When analyzing the mean values on the scales of coping strategies, we failed to identify differences between the groups with and without MACEs during the year after a PCI. Only when analyzing the frequency of using individual coping strategies could such differences be identified for confrontational coping; patients with MACEs more often preferred a moderate use of this strategy, and patients without MACEs preferred a rare use of this strategy. It is still difficult to understand what causes such differences in the data; perhaps the matter is in the ethnic characteristics of the patients, and the clinical indicators of the patients’ severity cannot be excluded. We cannot compare the latter, since such data were not provided in the previous study. Perhaps it can only be noted that confrontational coping also refers to a maladaptive way of responding to stressful situations, as well as the previously considered strategies of “Avoidance”, “Acceptance-humility”, and “Shifting responsibility”, which are characteristic of people with personality type D [8,38]. On the contrary, in patients with coronary artery disease in the study by Lv et al., the confrontation coping strategy was less common in people with personality type D [8].

The cognitive appraisal of a difficult life situation, although recognized as an integral part of an individual’s development of a strategy for coping with stress [39], is much less studied. With personality type D, such studies are generally sporadic [8,21,40], and these studies have obtained conflicting results. For example, Lv et al. [8] showed that the cognitive appraisal of a threat and challenge mediated a negative impact on the incidence of MACEs during the year after a PCI in CAD patients with type D personality. Another study showed that for elderly patients with personality type D, its negative impact on the ability to carry out self-help was mediated by the cognitive appraisal of one’s illness [40]. On the other hand, it was noted that the mediating effect of cognitive appraisal on the relationship between personality type D and smoothed cardiovascular stress reactivity was not significant [21]. In the cohort of patients examined by us, there was an association of personality type D with negative assessments of their life situations (feeling that their situations were not under control; orientation towards losses) (Sumin, in press). However, in the present study, the patients with MACEs had fewer cognitive appraisals such as strong emotions and future prospects compared to the patients without MACEs. The significance of our data, in our opinion, should be clarified in further studies.

The clarification of the behavioral pathophysiological mechanisms of the association between personality type D and prognosis after a PCI in this study is of clinical importance. Apparently, the evaluation of individual coping strategies should be integrated into the development of behavioral interventions in patients with personality type D. Considering the fact that there is a significant and constant relationship of maladaptive coping strategies in CAD patients with personality type D not only with the mental component of the quality of life and the development of depressive symptoms, but also with some behavioral habits (propensity for an unhealthy diet [41]), unsuccessful outcomes of cardiac rehabilitation programs [42], and prognosis [8], behavioral interventions that are specifically aimed at dysfunctional coping should be developed. In this direction of treatment, a possible impact on patients with coronary artery disease with personality type D is also seen to level its known adverse effect on prognosis. In addition, maladaptive coping strategies may be associated in different ways with type D personality and depressive symptoms in patients with coronary artery disease. This fact should be taken into account when developing future interventions aimed at preventing depression and mental disorders in this category of patients.

When analyzing the results of this study, the following limitations should be taken into account: First, patients assessed their psychological states using questionnaires, which could affect the adequacy of their self-assessments. Secondly, the study was conducted in only one center, so the possibility of replicating its results to other centers has not been proven. Third, the relatively small number of enrolled patients and the limited follow-up prevented the use of hard endpoints (death, non-fatal myocardial infarction, and non-fatal stroke) or the evaluation of individual MACEs only. Either the inclusion of additional patients or a longer follow-up of patients is necessary. A fourth limitation of this study can be considered potential problems with the validity and reliability of the cognitive assessment measures, since the CFI is less than 0.95 and the chi-squared test is statistically significant. Another limitation of this study is its insufficient power. Therefore, there may exist important differences in the population between the groups with and without MACEs that were not detected due to low power. The same applies to binary logistic regression. Also, we did not adjust for multiple testing when analyzing the results in Table 3 and Table 4, and this may have further reduced the power. It is possible to overcome these limitations of this study in further studies by increasing the number of patients included. Despite these limitations, this study can be considered a pilot study that will serve as a basis for the design of subsequent studies. Finally, statistically significant results were obtained only for personality type D as a dichotomous indicator, and not as a continuous value or as an effect of the interaction of two subscales. As shown earlier [43], this may overestimate the significance of the influence of personality type D on prognosis. However, most studies on type D personality provide data on its use as a dichotomous variable. This is quite consistent with common clinical practices that are commonly used in somatic medicine. For example, definitions such as diabetes or not, hypertension or not, and myocardial infarction or not are used (instead of presenting the glucose, blood pressure, or troponin as continuous values). Apparently, the use of a dichotomous approach to determine personality type D fits well into such a paradigm.

## 6. Conclusions

The present study showed that in patients with CAD with personality type D, in contrast to patients with non-type D personality, during the year after a PCI, MACEs and hospitalizations for cardiovascular diseases were more likely to develop. In the group with unfavorable prognoses, in addition to the more frequent occurrence of personality type D, there was a greater preference for a moderate use of the confrontational coping strategy. On cognitive appraisal, patients with MACEs had statistically significantly lower indicators of strong emotions and future prospects than patients without MACEs. The results of this study emphasize the need for individualized behavioral interventions for patients with coronary artery disease with manifestations of psychological distress. The possibility of improving prognosis through such interventions requires confirmation in further studies.

## Figures and Tables

**Figure 1 diagnostics-13-03374-f001:**
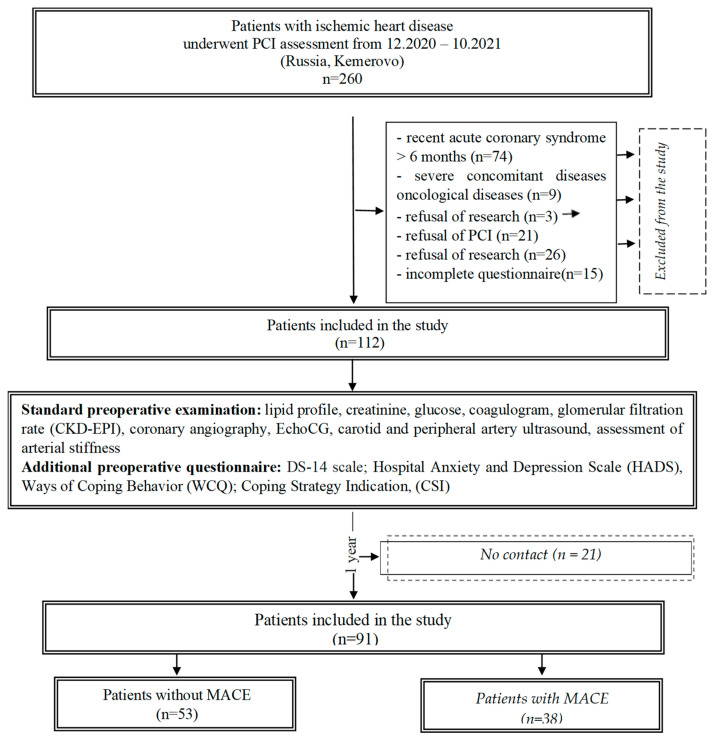
Flowchart of patient selection. MACE—major adverse cardiovascular event.

**Figure 2 diagnostics-13-03374-f002:**
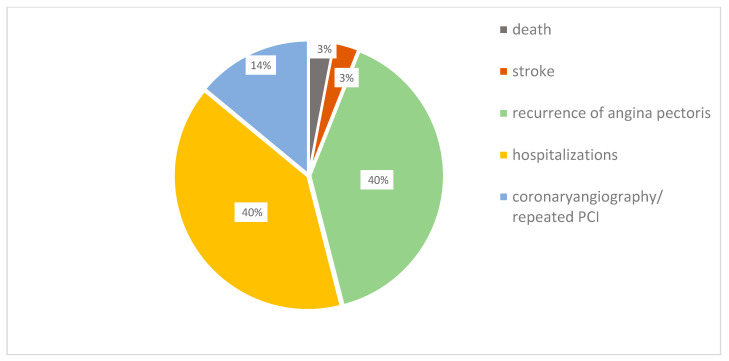
The structure of major adverse cardiovascular event after 1 year. Notes: PCI—percutaneous coronary intervention.

**Figure 3 diagnostics-13-03374-f003:**
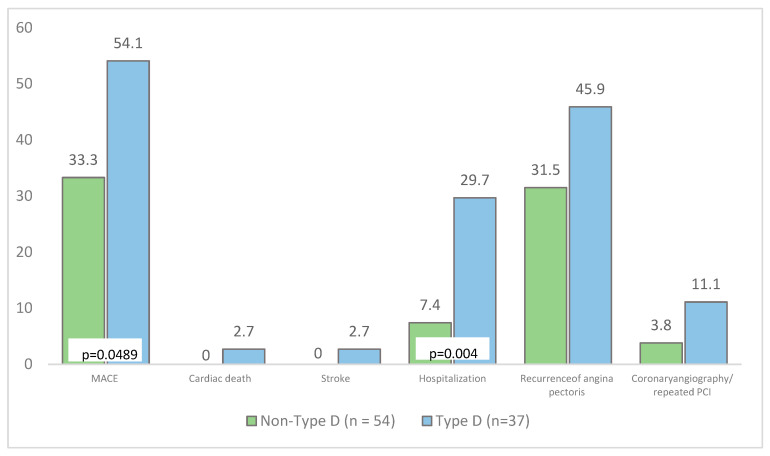
Analysis of MACEs and categorical type D personality status. Notes: MACE—major adverse cardiovascular event; PCI—percutaneous coronary intervention.

**Figure 4 diagnostics-13-03374-f004:**
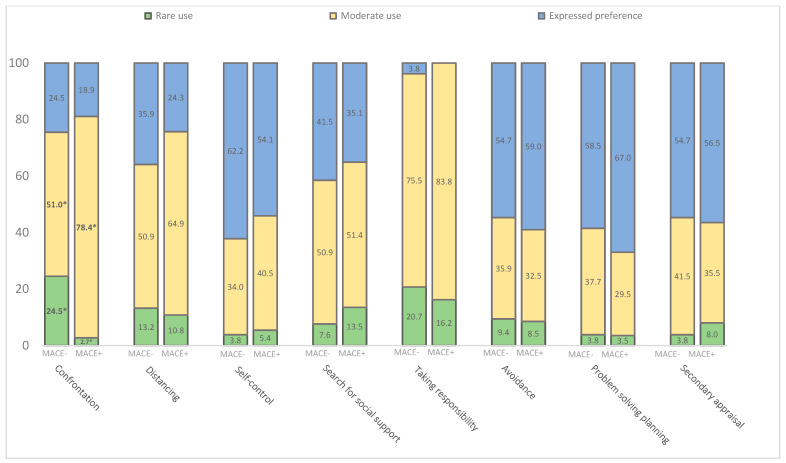
Distribution of coping strategies in CAD patients with and without MACEs 1 year after PCI (according to the WCQ questionnaire). * *p* < 0.001 compared MACE+ and MACE− groups.

**Figure 5 diagnostics-13-03374-f005:**
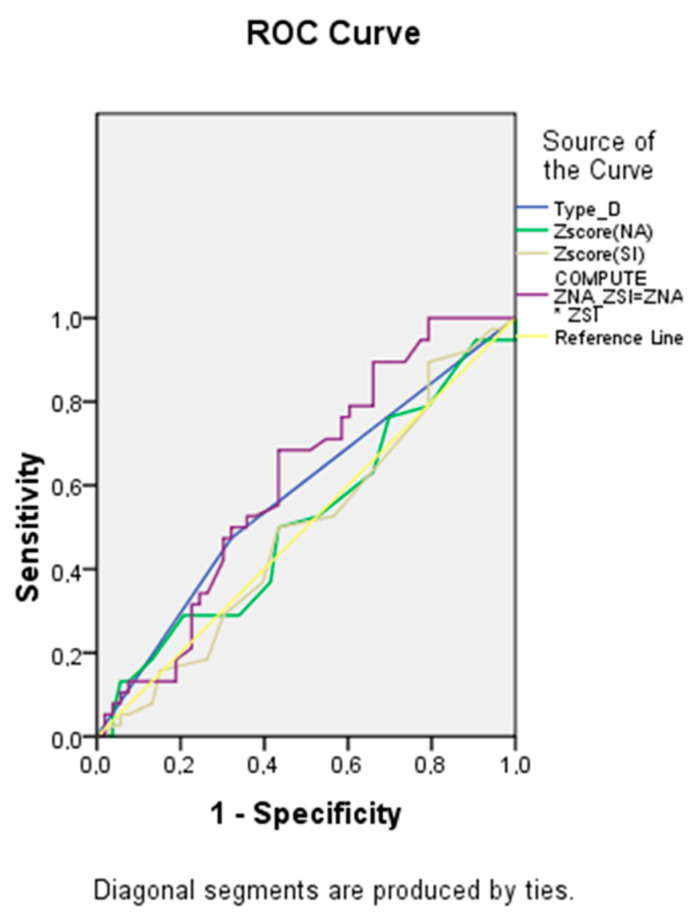
Receiver operating characteristic curve analysis. Performance of baseline parameters in discriminating unfavorable prognosis development.

**Table 1 diagnostics-13-03374-t001:** Baseline characteristics of patients with and without MACEs 1 year after PCI.

Variables	Group 1 (without MACEs) (*n* = 53)	Group 2 (with MACEs) (*n* = 38)	Z	*p*
Male (*n*, %)	31 (58.5)	20 (52.6)	-	0.578
Age, years	64.0 [58.0; 71.0]	65.5 [60.0; 69.0]	0.846	0.399
BMI, kg/m^2^	25.3 [21.4; 28.5]	25.4 [21.01; 27.4]	−0.189	0.98
Disability (*n*, %)	17 (32.1)	10 (26.3)	-	0.553
Working (*n*, %)	23 (43.4)	12 (32.43)	-	0.293
Current smoker (*n*, %)	22 (41.5)	16 (41.0)	-	0.929
Smoking experience, years	15.0 [0; 38.5]	20.0 [0; 35.0]	0.017	0.99
Hypertension, *n* (%)	45 (84.9)	31 (81.6)	-	0.521
Diabetes mellitus, *n* (%)	13 (24.5)	9 (23.7)	-	0.885
Stroke (*n*, %)	5 (9.4)	3 (7.9)	-	0.776
Myocardial infarction (*n*, %)	32 (60.4)	20 (52.6)	-	0.398
Previous CABG (*n*, %)	3 (5.7)	5 (13.2)	-	0.235
Carotid endarterectomy (*n*, %)	1 (1.9)	0	-	0.389
Angina class 0 (*n*, %)	11 (20.8)	5 (13.2)	-	0.327
I (*n*, %)	5 (9.4)	4 (10.5)	-	0.886
II (*n*, %)	33 (62.3)	29 (76.3)	-	0.193
III (*n*, %)	3 (5.7)	0	-	0.132
Heart failure class NYHA 0 (*n*, %)	1 (1.9)	1 (2.6)	-	0.811
I (*n*, %)	27 (50.9)	19 (50.0)	-	0.856
II (*n*, %)	24 (45.3)	17 (44.7)	-	0.869
III (*n*, %)	1 (1.9)	0	-	0.389
**Laboratory indicators**				
Total cholesterol, mmol/L	3.9 [3.5; 4.85]	4.25 [3.3; 5.4]	0.399	0.694
HDL cholesterol, mmol/L	1.04 [0.81; 3.2]	1.53 [1.01; 1.96]	1.937	0.061
LDL cholesterol, mmol/L	2.49 [1.81; 3.19]	2.7 [1.75; 3.8]	0.121	0.928
Triglyceride, mmol/L	1.4 [1.1; 1.7]	1.2 [0.9; 1.5]	−0.879	0.399
Creatinine, µmol/L	7.1 [5.9; 8.4]	6.65 [5.65; 8.75]	0.484	0.768
Glucose, mmol/L	90.0 [76.5; 104.0]	91.0 [80.0; 116.0]	0.636	0.632
**Coronarography**				
1—coronary artery disease, *n* (%)	40 (75.5)	26 (68.4)	-	0.457
2—coronary artery disease, *n* (%)	9 (17.7)	9 (23.7)	-	0.483
3—coronary artery disease, *n* (%)	4 (7.8)	3 (7.9)	-	0.992

Data are presented as the median and interquartile range (25th–75th percentile) or number (%), as indicated. Notes: BMI—body mass index; NYHA—New York Heart Association; CABG—coronary artery bypass grafting; HDL—high-density lipoproteins; LDL—low-density lipoproteins.

**Table 2 diagnostics-13-03374-t002:** Comparative analysis of the initial data of the questionnaire of patients with and without MACEs 1 year after PCI.

Variables	Group 1 (without MACEs) (*n* = 53)	Group 2 (with MACEs) (*n* = 38)	*p*
**DS-14**			
Type D personality, *n* (%)	17 (32.1)	20 (54.1)	0.049
Negative affectivity, points	10.32 ± 4.41	11.13 ± 4.92	0.361
Social inhibition, points	9.81 ± 3.91	10.36 ± 3.76	0.529
**HADS**			
Personal anxiety, points	6.07 ± 3.11	6.16 ± 3.17	0.961
Depression level, points	4.85 ± 3.43	4.97 ± 3.75	0.991
**Coping strategies (WCQ)**			
Confrontation, points	9.6 ± 4.0	10.91 ± 2.67	0.187
Distancing, points	10.94 ± 3.75	10.83 ± 3.22	0.665
Self-control, points	12.85 ± 3.58	12.75 ± 3.85	0.908
Search for social support, points	11.94 ± 4.17	11.73 ± 3.63	0.634
Taking responsibility, points	8.53 ± 2.81	8.51 ± 2.28	0.996
Avoidance, points	12.26 ± 4.18	12.43 ± 3.89	0.738
Problem solving planning, points	13.01 ± 3.34	13.91 ± 2.96	0.219
Secondary appraisal, points	12.91 ± 3.87	13.37 ± 3.89	0.478

Note: M (SD) = mean (standard deviation).

**Table 3 diagnostics-13-03374-t003:** Comparative analysis of data from the “Appraisal Criteria of the Life Situation Difficulty” questionnaire for groups of patients with the presence/absence of MACEs.

Variables	Group 1 (without MACEs) (*n* = 53)	Group 2 (with MACEs) (*n* = 38)	*p*
Common signs of difficult life situations, points	16.1 ± 4.9	14.94 ± 5.06	0.108
Lack of control of the situation, points	10.38 ± 4.49	10.86 ± 4.84	0.746
Incomprehensibility of the situation, points	16.0 ± 7.01	14.42 ± 6.75	0.293
The need for a quick and active response, points	16.2 ± 4.96	15.81 ± 4.61	0.688
Difficulties in making a decision, points	14.9 ± 4.71	13.39 ± 5.21	0.108
Difficulties in predicting the situation, points	10.54 ± 4.41	9.36 ± 4.22	0.234
Strong emotions, points	14.62 ± 4.83	11.92 ± 5.32	0.005
Future perspective, points	13.21 ± 3.41	11.36 ± 3.81	0.015

**Table 4 diagnostics-13-03374-t004:** Comparative analysis of the “Types of orientations in difficult situations” questionnaire data for groups of patients with the presence/absence of MACEs.

Variables	Group 1 (without MACEs) (*n* = 53)	Group 2 (with MACEs) (*n* = 38)	*p*
Drive, points	22.27 ± 5.85	22.73 ± 5.93	0.956
Thoroughness, points	11.75 ± 2.27	12.24 ± 2.84	0.409
Threat alert, points	12.23 ± 2.76	11.84 ± 3.19	0.554
Opportunity orientation, points	17.91 ± 2.67	18.54 ± 4.39	0.743
Obstacle orientation, points	12.34 ± 2.58	12.08 ± 3.23	0.603
Rejection, points	15.98 ± 4.59	16.11 ± 5.77	0.408
Inaction, points	14.35 ± 3.73	15.49 ± 4.48	0.094
Insouciance, points	11.58 ± 3.23	11.49 ± 4.02	0.844

**Table 5 diagnostics-13-03374-t005:** Results of binary logistic regression (forward LR method): association of factors with the risk of unfavorable prognosis development.

		B	S.E.	Wald	df	Sig.	Exp (B)	95% CI for EXP (B)	
								Lower	Upper
Step 1	Confrontation (WSQ), moderate use	1.253	0.542	5.344	1	0.021	3.500	1.233	9.399
	Constant	−1.099	0.436	6.336	1	0.012	0.333		
Step 2	Type_D	1.075	0.552	3.797	1	0.051	2.931	1.064	9.970
	Confrontation (WSQ), moderate use	1.262	0.560	5.082	1	0.024	3.532	1.283	9.842
	Constant	−1.505	0.508	8.782	1	0.003	0.222		
Step 3	Type_D	1.291	0.591	4.766	1	0.029	3.636	1.265	10.894
	Confrontation (WSQ), moderate use	1.641	0.649	6.400	1	0.011	5.160	1.472	14.823
	Strong emotions (ACLSD)	−0.132	0.062	4.495	1	0.034	0.876	0.720	0.959
	Constant	−0.150	0.896	0.028	1	0.867	0.860		
Step 4	Type_D	1.381	0.617	5.011	1	0.025	3.978	1.324	12.890
	Confrontation (WSQ), moderate use	1.758	0.669	6.900	1	0.009	5.801	1.689	20.297
	Self-control (WSQ), moderate use	1.343	0.651	4.253	1	0.039	3.829	1.299	11.100
	Strong emotions (ACLSD)	−0.132	0.064	4.218	1	0.040	0.876	0.787	0.978
	Constant	−0.715	0.977	0.536	1	0.464	0.489		

**Table 6 diagnostics-13-03374-t006:** Results of binary logistic regression (enter method): association of factors (Type D, NA, SI, and ZNA × ZSI) with the risk of the unfavorable prognosis development.

	B	S.E.	Wald	df	Sig.	Exp(B)	95% CI for EXP (B)	
							Lower	Upper
Type_D	1.297	0.688	3.551	1	0.060	3.660	0.949	14.107
NA	−0.042	0.068	0.381	1	0.537	0.959	0.839	1.096
SI	−0.105	0.092	1.294	1	0.255	0.901	0.752	1.079
ZNA_ZSI	0.219	0.208	1.108	1	0.293	1.245	0.828	1.874
Constant	0.497	0.799	0.387	1	0534	1.644		

**Table 7 diagnostics-13-03374-t007:** Receiver operating characteristic curve analysis. Performance of baseline parameters in discriminating unfavorable prognosis development. Area under the curve.

Test Result Variable(s)				Asymptotic 95% Confidence Interval	
	Area	Std. Error	Asymptotic Sig.	Lower Bound	Upper Bound
Type D	0.576	0.061	0.215	0.456	0.697
Zscore NA	0.514	0.062	0.815	0.393	0.636
Zscore SI	0.495	0.061	0.936	0.375	0.615
ZNA × ZSI	0.613	0.059	0.066	0.499	0.728

## Data Availability

The datasets used and/or analyzed during the current study available from the corresponding author on reasonable request.

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
