# Peer review of "Pathogenetic Mechanisms Underlying Major Adverse Cardiac Events in Personality Type D Patients after Percutaneous Coronary Intervention: The Roles of Cognitive Appraisal and Coping Strategies"

_diagnostics, 2023, doi:10.3390/diagnostics13213374_

Round 1
Reviewer 1 Report
Comments and Suggestions for Authors
The current manuscript concerns an investigation of the relation between various psychological factors and the occurrence of MACE in a sample of 111 CAD patients who have received PCI. I think this study is a potentially relevant contribution to the literature and although I think it is well conducted, it can still be improved by considering the points raised below:
· Page 3: Near the end of the introduction various coping strategies are mentioned but not defined. To increase the clarity of the introduction the authors could shortly introduce the coping styles relevant to the current study.
· Page 4: Various details on the measurement instruments are missing. For instance, their reliability estimates in the current data, an example of a question including the response options, the number of HADS items per construct.
· Page 5: The model fit for the two CFAs on the cognitive appraisal measures is not sufficient because the CFI is smaller than 0.95 and the chi-square test statistically significant. This raises concerns regarding the validity of the results reported in Tables 3 and 4, because the construct validity of these 8 + 5 variables has not been adequately established. Also, please report reliability estimates for each subscale. Low reliability increases the risk that any reported non-significant differences between MACE and no MACE in Table 3 and 4 are due to measurement error obfuscating true population differences. Potential issues with the validity and reliability of the cognitive appraisal measures could be reported as a limitation.
· Page 6: For the logistic regression the authors report a set of questionnaires to be included in the model, but I think it would be more accurate to report the actual variables. For instance, instead of reporting that the DS14 questionnaire was included, the authors could report the specific included variables that are used to analyze Type D personality in this model.
· Page 6: A total of 21 patients did not respond at follow-up, which means a dropout rate of 18.75%. If the authors seek to generalize their findings to the entire population of patients undergoing PCI then it would be important to present a Table in the supplement showing the differences between the dropouts and completers in terms of baseline characteristics.
· Table 1: The MACE and no-MACE groups have been compared on various patient characteristics. However, these analyses were likely underpowered. For instance, a t-test comparing two groups of 53 and 38 participants only has sufficient power (0.80) to detect large standardized mean differences between the groups (d = 0.60). So the two groups are only 'comparable in all respects' in terms or large mean differences. For mean differences smaller than 0.60 the study is underpowered, so it remains uncertain whether these characteristics differ in the population. Please report as a limitation that there may exist important differences in the population between these groups that could not be detected due to low power. The same applies to the binary logistic regression.
· Table 5 could be further improved. It is currently quite difficult to see which variables belong to the same model. The authors could consider separating the various modeling steps with a separator line. Please also report the 95% confidence intervals of the odds ratio estimates.
· No specific hypotheses have been reported for the relation between MACE and the various coping strategies or cognitive appraisals. Therefore, there are many p-values that can be used in support of the question whether these constructs are related to MACE. To prevent inflated false positive rates in such exploratory analyses, it is common to apply a correction for multiple testing to the analyses reported in Tables 3 and 4. Yet considering the small sample size this may further decrease power, so I recommend reporting this as a limitation.
· As a sensitivity analysis it would be interesting to see the results of a binary logistic regression analysis (without variable selection) including only control variables and the dimensional Type D variables (NA, SI, NA*SI). The currently reported significant Type D effect based on a dichotomous Type D variable can either be due to a main effect of NA, or a main effect of SI, or main effects for both, or a synergy between them. So currently it remains unclear whether it is really Type D, or merely NA or SI that is related to MACE.
Author Response
The current manuscript concerns an investigation of the relation between various psychological factors and the occurrence of MACE in a sample of 111 CAD patients who have received PCI. I think this study is a potentially relevant contribution to the literature and although I think it is well conducted, it can still be improved by considering the points raised below:
Thanks to the reviewer for his high assessment of our manuscript and useful comments, which we took into account when correcting the manuscript and which, as we hope, allowed us to improve it.
- Page 3: Near the end of the introduction various coping strategies are mentioned but not defined. To increase the clarity of the introduction the authors could shortly introduce the coping styles relevant to the current study.
We have made an addition to the Introduction section, adding the following phrase:
«For example, among the adequate strategies for overcoming stress are "Planful Problem-Solving", "Self-Control", "Positive Appraisal", and "Accepting Responsibility". The following coping strategies are less adequate - "Confrontative Coping", "Seeking Social Support", and especially "Dis-tancing and Escape/Avoidance" [24]».
- Page 4: Various details on the measurement instruments are missing. For instance, their reliability estimates in the current data, an example of a question including the response options, the number of HADS items per construct.
We have added information about the DS-14 and HADS questionnaires. We did not provide information in the article about examples of questions and answer options, since this journal is not a specialized psychological journal, and these questionnaires are widely presented in the scientific literature (including for cardiac patients). If the reviewer considers it necessary, then it is possible to place these questionnaires as an addition to the text of the article.
- Page 5: The model fit for the two CFAs on the cognitive appraisal measures is not sufficient because the CFI is smaller than 0.95 and the chi-square test statistically significant. This raises concerns regarding the validity of the results reported in Tables 3 and 4, because the construct validity of these 8 + 5 variables has not been adequately established. Also, please report reliability estimates for each subscale. Low reliability increases the risk that any reported non-significant differences between MACE and no MACE in Table 3 and 4 are due to measurement error obfuscating true population differences. Potential issues with the validity and reliability of the cognitive appraisal measures could be reported as a limitation.
Thanks for the important clarification. In connection with your comment, we have supplemented the Study Limitation section with the following phrase:
“Fourth, a limitation of the study can be considered potential problems with the validity and reliability of the cognitive assessment measures, since the CFI is less than 0.95 and the chi-square test is statistically significant. Another limitation of the study is its unsufficient power. Therefore, there may exist important differences in the population between the groups with and without MACE that were not detected due to low power. The same applies to binary logistic regression. Also we did not adjust for multiple testing when analyzing the results in Tables 3 and 4, as this may have further reduced power. It is possible to overcome these limitations of the study in further studies by increasing the number of patients included. Despite these limitations, this study can be considered a pilot study that will serve as a basis for the design of subsequent studies.”
- Page 6: For the logistic regression the authors report a set of questionnaires to be included in the model, but I think it would be more accurate to report the actual variables. For instance, instead of reporting that the DS14 questionnaire was included, the authors could report the specific included variables that are used to analyze Type D personality in this model.
Thanks for the suggestion, we have included specific data from the DS 14 questionnaire in the logistic regression model described.
- Page 6: A total of 21 patients did not respond at follow-up, which means a dropout rate of 18.75%. If the authors seek to generalize their findings to the entire population of patients undergoing PCI then it would be important to present a Table in the supplement showing the differences between the dropouts and completers in terms of baseline characteristics.
Thank you for your comment. We additionally compared the patients who dropped out of the study and those presented in the article; the results of this comparison showed the comparability of these groups in terms of initial indicators. This table is given in the Supplement.
- Table 1: The MACE and no-MACE groups have been compared on various patient characteristics. However, these analyses were likely underpowered. For instance, a t-test comparing two groups of 53 and 38 participants only has sufficient power (0.80) to detect large standardized mean differences between the groups (d = 0.60). So the two groups are only 'comparable in all respects' in terms or large mean differences. For mean differences smaller than 0.60 the study is underpowered, so it remains uncertain whether these characteristics differ in the population. Please report as a limitation that there may exist important differences in the population between these groups that could not be detected due to low power. The same applies to the binary logistic regression.
Thanks for the important clarification. In connection with your comment, we have supplemented the Study Limitation section with the following phrase:
“Fourth, a limitation of the study can be considered potential problems with the validity and reliability of the cognitive assessment measures, since the CFI is less than 0.95 and the chi-square test is statistically significant. Another limitation of the study is its unsufficient power. Therefore, there may exist important differences in the population between the groups with and without MACE that were not detected due to low power. The same ap-plies to binary logistic regression. Also we did not adjust for multiple testing when ana-lyzing the results in Tables 3 and 4, as this may have further reduced power. It is possible to overcome these limitations of the study in further studies by increasing the number of patients included. Despite these limitations, this study can be considered a pilot study that will serve as a basis for the design of subsequent studies.”
- Table 5 could be further improved. It is currently quite difficult to see which variables belong to the same model. The authors could consider separating the various modeling steps with a separator line. Please also report the 95% confidence intervals of the odds ratio estimates.
We have adjusted Table 5 and added dividing lines, and 95% confidence intervals
- No specific hypotheses have been reported for the relation between MACE and the various coping strategies or cognitive appraisals. Therefore, there are many p-values that can be used in support of the question whether these constructs are related to MACE. To prevent inflated false positive rates in such exploratory analyses, it is common to apply a correction for multiple testing to the analyses reported in Tables 3 and 4. Yet considering the small sample size this may further decrease power, so I recommend reporting this as a limitation.
Thanks for the important clarification. In connection with your comment, we have supplemented the Study Limitation section with the following phrase:
“Fourth, a limitation of the study can be considered potential problems with the validity and reliability of the cognitive assessment measures, since the CFI is less than 0.95 and the chi-square test is statistically significant. Another limitation of the study is its unsufficient power. Therefore, there may exist important differences in the population between the groups with and without MACE that were not detected due to low power. The same ap-plies to binary logistic regression. Also we did not adjust for multiple testing when ana-lyzing the results in Tables 3 and 4, as this may have further reduced power. It is possible to overcome these limitations of the study in further studies by increasing the number of patients included. Despite these limitations, this study can be considered a pilot study that will serve as a basis for the design of subsequent studies.”
- As a sensitivity analysis it would be interesting to see the results of a binary logistic regression analysis (without variable selection) including only control variables and the dimensional Type D variables (NA, SI, NA*SI). The currently reported significant Type D effect based on a dichotomous Type D variable can either be due to a main effect of NA, or a main effect of SI, or main effects for both, or a synergy between them. So currently it remains unclear whether it is really Type D, or merely NA or SI that is related to MACE.
We additionally conducted a binary logistic regression analysis (without variable selection) including only control variables and type D dimensional variables (NA, SI, NA*SI). We present this in an additional table (see below). It shows that the effect of type D on the development of MACE, based on the dichotomous variable of type D, was more pronounced than the main effect of NA, or the main effect of SI, or the synergy between them.
|
|
B |
S.E. |
Wald |
df |
Sig. |
Exp(B) |
95% C.I.for EXP(B) |
|
|
|
Lower |
Upper |
||||||
Step 1a |
Type_D |
1,297 |
0,688 |
3,551 |
1 |
0,060 |
3,660 |
0,949 |
14,107 |
NA |
-0,042 |
0,068 |
0,381 |
1 |
0,537 |
0,959 |
0,839 |
1,096 |
|
SI |
-0,105 |
0,092 |
1,294 |
1 |
0,255 |
0,901 |
0,752 |
1,079 |
|
ZNA_ZSI |
0,219 |
0,208 |
1,108 |
1 |
0,293 |
1,245 |
0,828 |
1,874 |
|
Constant |
0,497 |
0,799 |
0,387 |
1 |
0,534 |
1,644 |
|
|
Reviewer 2 Report
Comments and Suggestions for Authors
The manuscript presents a research proposal of interest, but requires improvements that need to be modified, among them are:
- It is suggested that the Material and Method section have as subsections: Participants, Instruments, Procedure and Data Analysis.
- The authors indicate that for some analyzes they use parametric tests (Student's t) and for other analyzes non-parametric tests (Mann-Whitney), it is suggested to unify the type of tests to be performed and whether the distribution of the scores follows a non-parametric distribution. normal must use non-parametric tests in all cases.
- The effect size of the tests carried out and the statistical power are not reported; the authors must include this information.
- An update of the reviewed sources is necessary, both for the introduction and review of the theoretical framework, and for the discussion of the results, where relevant international studies are provided, especially from the last 3 years.
- 14 references from the last 3 years are presented, less than 30% of the total references. It is suggested to increase the number of studies from recent years, in order to increase the thematic current index of the study.
Author Response
The manuscript presents a research proposal of interest, but requires improvements that need to be modified, among them are:
- It is suggested that the Material and Method section have as subsections: Participants, Instruments, Procedure and Data Analysis.
Dear reviewer, your proposal is completely justified and requires consideration. However, we used other section titles - 2. Materials and Methods, 2.1. Patients and procedures, 3. Measures, 3.1. Type D personality, 3.2. Hospital Anxiety and Depression Scale (HADS) personality, 3.3. Coping style, 3.4. Cognitive appraisal, 3.5. Follow-up observation, 3.6. Statistical analyses. In our opinion, such sections are quite adequate for our article; perhaps they will suit the respected reviewer.
- The authors indicate that for some analyzes they use parametric tests (Student's t) and for other analyzes non-parametric tests (Mann-Whitney), it is suggested to unify the type of tests to be performed and whether the distribution of the scores follows a non-parametric distribution. normal must use non-parametric tests in all cases.
We assessed the distribution of quantitative variables and checked for normality using the Kolmogorov-Smirnov test. With a normal distribution, the Student's t-test was used when comparing two groups; with a nonparametric distribution, the Mann-Whitney test was used.
- The effect size of the tests carried out and the statistical power are not reported; the authors must include this information.
We have added information about the power of the study to the article. Since the power of the study was not high enough, we noted this in the Study Limitations section.
- An update of the reviewed sources is necessary, both for the introduction and review of the theoretical framework, and for the discussion of the results, where relevant international studies are provided, especially from the last 3 years.
We have adjusted the list of references and supplemented it with more recent publications.
- 14 references from the last 3 years are presented, less than 30% of the total references. It is suggested to increase the number of studies from recent years, in order to increase the thematic current index of the study.
We have adjusted the list of references and supplemented it with more recent publications.
Round 2
Reviewer 1 Report
Comments and Suggestions for Authors
The authors have adequately handled most of my previous comments in this revision. I only have a few more comments:
* Good to see that the authors have performed an additional analysis including the dimensional Type D variables in a logistic regression. However, these estimates are difficult to interpret because both the categorical Type D variable and the continuous Type D variables (zNA, zSI, zNA*zSI) appear to be included in a single model. The continuous variables may not reach significance because the categorical Type D variable is already included. I recommend fitting this continuous Type D model again but without the categorical Type D variable. Then I suggest to include the resulting Table in the manuscript and discuss the findings at the end of the results section as a sensitivity analysis.
* Line 146: "even points 2, 4.6; 8 ..." can be rewritten as "even items 2, 4, 6, 8, ..."
Author Response
The authors have adequately handled most of my previous comments in this revision. I only have a few more comments:
Thank you for your high appreciation of the work we did to correct the manuscript.
* Good to see that the authors have performed an additional analysis including the dimensional Type D variables in a logistic regression. However, these estimates are difficult to interpret because both the categorical Type D variable and the continuous Type D variables (zNA, zSI, zNA*zSI) appear to be included in a single model. The continuous variables may not reach significance because the categorical Type D variable is already included. I recommend fitting this continuous Type D model again but without the categorical Type D variable. Then I suggest to include the resulting Table in the manuscript and discuss the findings at the end of the results section as a sensitivity analysis.
We added Table 6, which presented the results of a binary logistic regression model that included various personality type D construct factors. These data confirmed the results presented in Table 5, where only Type D as a dichotomous variable had an independent association with MACE.
* Line 146: "even points 2, 4.6; 8 ..." can be rewritten as "even items 2, 4, 6, 8, ..."
We have made corrections to the text of the manuscript.
Reviewer 2 Report
Comments and Suggestions for Authors
I consider that the authors have improved the quality of the manuscript with the changes made, however they do not have to make some minor changes:
- Improve and review the format of the tables, they present a "copy and paste" of the program used for the analyzes without adapting it to the format of the article.
- Check for some typographical errors when presenting the items that are part of each subscale of the instruments used (no spaces between commas, etc.).
- It is necessary to present the results in the most didactic format, easier to interpret, following the recommendations for editing them.
- The authors still do not report the effect size in the Mann-Whitney U tests, they report the statistical significance (p value) but not the effect size.
Author Response
I consider that the authors have improved the quality of the manuscript with the changes made, however they do not have to make some minor changes:
Thank you for your high appreciation of the work we did to correct the manuscript.
- Improve and review the format of the tables, they present a "copy and paste" of the program used for the analyzes without adapting it to the format of the article.
We have made corrections to the text of the manuscript.
- Check for some typographical errors when presenting the items that are part of each subscale of the instruments used (no spaces between commas, etc.).
We have made corrections to the text of the manuscript.
- It is necessary to present the results in the most didactic format, easier to interpret, following the recommendations for editing them.
We have made corrections to the text of the manuscript.
- The authors still do not report the effect size in the Mann-Whitney U tests, they report the statistical significance (p value) but not the effect size.
We have added to Table 1 the values of Z, which is used for the effect size in the Mann-Whitney U tests